There are amendments to this paper

# Combined burden and functional impact tests for cancer driver discovery using DriverPower

Shimin Shuai [1,2]*, PCAWG Drivers and Functional Interpretation Working Group, Steven Gallinger[3,4], Lincoln D. Stein[1,2]* & PCAWG Consortium

The discovery of driver mutations is one of the key motivations for cancer genome sequencing. *Here, as part of the ICGC/TCGA Pan-Cancer Analysis of Whole Genomes (PCAWG) Consortium*, which aggregated whole genome sequencing data from 2658 cancers across 38 tumour types, we describe DriverPower, a software package that uses mutational burden and functional impact evidence to identify driver mutations in coding and non-coding sites within cancer whole genomes. Using a total of 1373 genomic features derived from public sources, DriverPower's background mutation model explains up to 93% of the regional variance in the mutation rate across multiple tumour types. By incorporating functional impact scores, we are able to further increase the accuracy of driver discovery. Testing across a collection of 2583 cancer genomes from the PCAWG project, DriverPower identifies 217 coding and 95 non-coding driver candidates. Comparing to six published methods used by the PCAWG Drivers and Functional Interpretation Working Group, DriverPower has the highest F1 score for both coding and non-coding driver discovery. This demonstrates that DriverPower is an effective framework for computational driver discovery.

[1] Department of Molecular Genetics, University of Toronto, Toronto, ON, Canada M5S 1A8. [2] Computational Biology Program, Ontario Institute for Cancer Research, Toronto, ON, Canada M5G 0A3. [3] Division of General Surgery, Toronto General Hospital, Toronto, ON, Canada M5G 2C4. [4] Lunenfeld-Tanenbaum Research Institute, Mount Sinai Hospital, Toronto, ON, Canada M5G 1X5. PCAWG Drivers and Functional Interpretation Working Group authors and their affiliations appear at the end of the paper. PCAWG Consortium members and their affiliations appear online. *email: shimin.shuai@mail.utoronto.ca; lincoln.stein@gmail.com

Cancer drivers are somatic genetic alterations that confer selective advantages to tumour cells[1,2]. Identification of cancer drivers is a crucial yet challenging task in cancer genomics research[3,4]. There are multiple challenges. First, driver mutations generally account for only a small fraction of the somatic variations found in a typical tumour, the rest being innocent bystander 'passenger' mutations[5]. Second, there is substantial intra- and inter-tumoural heterogeneity in most cancers[6]. Both across different tumour types and across different genomic regions within the same tumour, the background mutation rate (BMR) can vary over several orders of magnitude.

The advent of large-scale cancer whole-genome sequencing (WGS) data has made it possible to explore the role of driver events in non-coding regions. However, identifying non-coding driver events in WGS creates new challenges. First, although the functional impact of somatic mutations in the coding regions of genes is fairly straightforward to predict, much less is known about the effect of mutations on non-coding regions of the genome. Second, only ~1% of somatic mutations detected in PCAWG WGS data are exonic, adding substantially more mutations and regions to be tested and demanding more careful control of type I and type II errors than WGS. At present, only a limited number of non-coding drivers are known, the primary examples being the *TERT* promoter for multiple tumour types and the *TAL1* enhancer for T-cell acute lymphoblastic leukaemia[7,8].

Most state-of-the-art methods identify drivers by detecting signals of positive selection either through mutational burden tests, which compare the rate of mutations observed in a region of the genome to what is expected from the BMR, or functional impact tests, which identify putative driver mutations based on a higher-than-expected rate of changes that are predicted to alter the function of genomic elements[3,6]. Mutational burden tests work best for calling frequently recurrent driver events and perform poorly when applied to rare driver events. In contrast, functional impact tests fail to find drivers in genomic elements that are poorly understood or annotated.

To maximise accuracy, we combined the two mutation significance testing methods to develop DriverPower (Fig. 1a), a framework for identification of coding and non-coding cancer drivers using mutational burden and functional impact scores. We first present the DriverPower method and describe the candidate driver mutations identified by applying the method to the ICGC/TCGA Pan-Cancer Analysis of Whole Genomes (PCAWG) data set. The PCAWG Consortium aggregated WGS data from 2658 cancers across 38 tumour types generated by the ICGC and TCGA projects. These sequencing data were reanalysed with standardised, high-accuracy pipelines to align to the human genome (reference build hs37d5) and identify germline variants and somatically acquired mutations, as described in ref. [9]. Then we show that DriverPower outperforms several published methods for both coding and non-coding driver discovery and discuss some novel candidates identified by DriverPower.

## Results

### Features predictive of BMR.
To evaluate DriverPower, we took WGS somatic variant data derived from 2583 high-quality donors from the PCAWG project[9]. After removing hypermutated samples, 2514 donors with 24,715,214 somatic single nucleotide variants (SNV) and small indels were used for driver element identification. We analysed these data both as a single pan-cancer data set, as well as a series of 29 tumour type-specific cohorts (Supplementary Data 1).

Among all tumour cohorts, we observed substantial variability in the observed mutation rate at the tissue, donor and locus levels (Supplementary Figs. 1 and 2). Accurate driver detection requires an accurate estimate of BMR across the tumour genome, taking into account the extensive variability among tumour types, donors and genomic regions. DriverPower tackles this issue by modelling the BMR using numerous genomic features that co-vary with the localised BMR. We collected 1373 features from three public data portals (Supplementary Data 2): the ROADMAP Epigenomics project, the ENCODE project and the UCSC genome browser[10–12]. These features covered seven main categories: conservation, DNA accessibility, epigenomic marks, nucleotide contents, replication timing, RNA expression and genome compartments. As expected, we found extensive multicollinearity among features. Most features (1368/1373) are significantly (Spearman's rho test $q < 0.1$) correlated with pan-cancer genome-wide mutation rates (Supplementary Fig. 3).

### BMR model.
We investigated two algorithms for modelling the BMR based on genomic features. The first algorithm was randomised lasso followed by binomial generalised linear model (GLM). The alternative algorithm was the gradient boosting machine (GBM), which is a non-linear and non-parametric tree ensemble algorithm[13]. To evaluate both BMR modelling algorithms, we made non-overlapped 1 megabase pair (Mbp) autosomal elements ($n = 2521$) as well as training genomic elements ($n = 867,266$) by sampling genomic coordinates randomly. The number of mutations per element was then predicted with five-fold cross validation (CV).

When evaluated using 1-Mbp autosomal elements, we found that both algorithms could accurately predict the BMR (Supplementary Figs. 4 and 5). In high mutational burden tumour cohorts, we observed essentially no difference between two algorithms, however GBM consistently outperformed GLM when applied to low mutational burden tumour cohorts (Supplementary Fig. 6). When evaluated on the training element set, in which the size of element varies from 100 bp to 1 Mbp, the prediction accuracy drops due to higher BMR variability, especially for low mutational burden tumour cohorts such as Myeloid-MPN and CNS-PiloAstro (Supplementary Fig. 6). However, for large cohort such as the pan-cancer set ($N = 2253$), ~ 93% of the mutation rate variance on the training set is explained by either model (Fig. 1b). The model still shows excellent performance when applied to the test element set, explaining 83% of the mutation rate variance on the pan-cancer cohort (Fig. 1c).

Both the randomised lasso algorithm and the GBM can be used to rank feature importance in different ways. Feature selection ranking from both methods confirmed that H3K9me3 (associated with heterochromatin), replication timing and H3K27ac (or its antagonistic histone mark H3K27me3) are the most important groups of predictors for BMR (Supplementary Fig. 7 and Supplementary Data 2)[14]. Consistent with previous results, we found that features from tumour cell lines with similar cell-of-origin to the primary tumour type are frequently selected[15]. For example, replication timing from liver cancer cell line HepG2 was selected as a feature for the BMR in hepatocellular carcinoma (Liver-HCC), whereas replication timing in MCF7 (breast cancer) and SK-N-SH (neuroblastoma) were selected for breast adenocarcinoma (Breast-AdenoCA) and glioblastoma (CNS-GBM), respectively (Supplementary Fig. 8).

### Functional adjustment.
In most burden-based methods, mutations are equally weighted. However, not all mutations have the same functional consequences. To incorporate functional consequence information, DriverPower implements a posterior functional adjustment. The functional adjustment step up-weights mutations with high predicted functional impact.

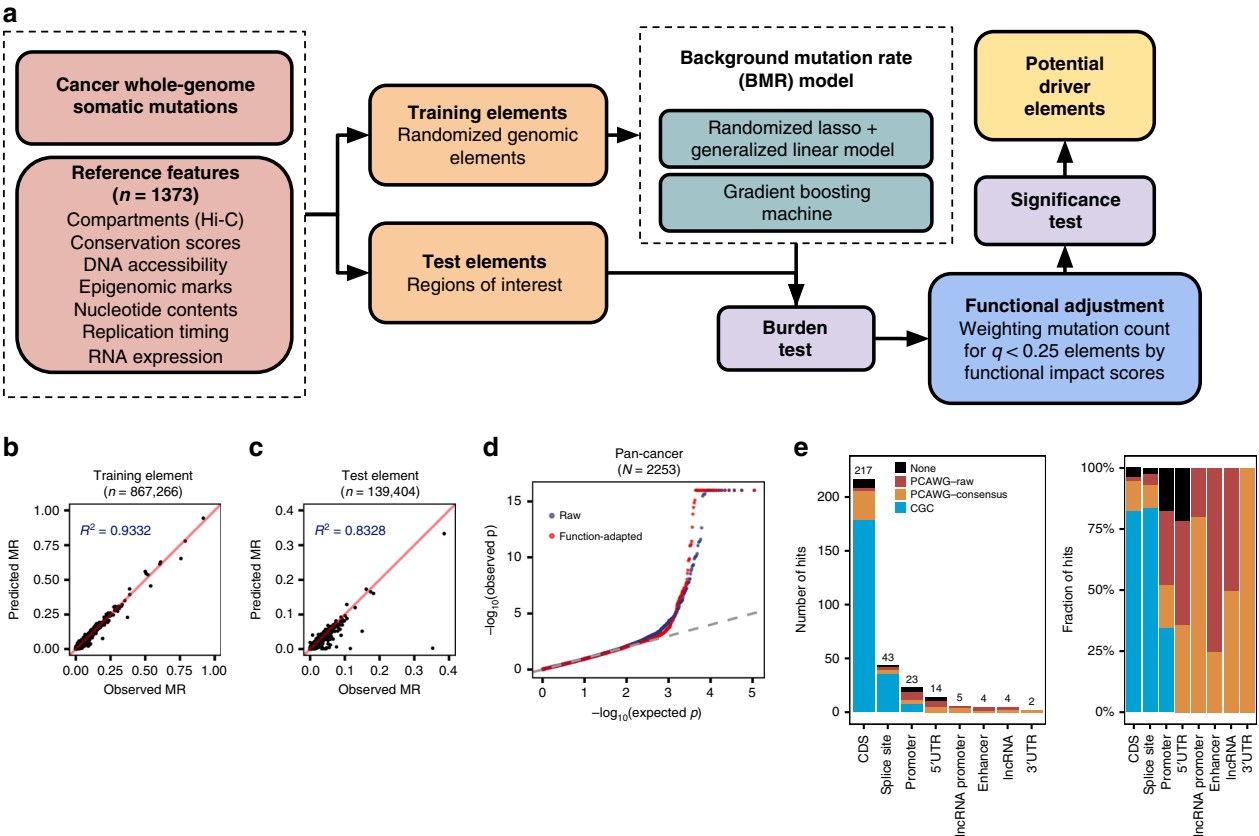

**Fig. 1 Summary of method and results. a** DriverPower overview. **b**, **c** For the training and test element sets, comparison of the predicted (Y axis) and observed (X axis) mutation rate in the pan-cancer cohort. **d** The raw and function-adapted p value quantile-quantile (QQ)-plot for all test elements in the pan-cancer cohort. Function-adapted p values are p values with the incorporation of functional impact scores. **e** Number and fraction of non-coding driver candidates called by DriverPower contained within three reference driver sets (CGC, PCAWG-consensus or PCAWG-raw). For each element type, the number of candidates is also shown above the bar.

Althyough the DriverPower framework can potentially work with any functional scoring scheme, in the current implementation we measured the functional impact using four published scoring schemes: the CADD[16], DANN[17], EIGEN[18] and LINSIGHT[19] scores. Although different training data, assumptions and algorithms are used by different scores, we found those scores to be consistent at the element level (Supplementary Fig. 9). We used the average weight of all four scores in the remainder of the manuscript unless otherwise specified.

**Candidate driver event discovery**. To evaluate the DriverPower algorithm, we first employed three simulated variant sets generated by the PCAWG Drivers and Functional Interpretation Working Group (PDFIWG) to examine type I and type II errors. We expected to identify no drivers as all three simulated data sets are reshuffles of observed mutations. In general, we observed no inflation or deflation in simulations and only eight significant hits (DriverPower $q < 0.1$) were identified in ~ 11 M statistical tests (Supplementary Fig. 10). We then used the observed PCAWG data set to discover drivers within multiple coding and non-coding element sets identified by the PDFIWG, spanning 3.7% (~ 113 Mbp) of the human genome.

We benchmarked our results against reference driver element sets and driver candidates called by six other published methods. Among the six methods, ExInAtor[20], ncdDetect[21] and LARVA[22] use only mutational burden information; oncodriveFML[23] uses only functional biases; whereas MutSig[24] and ActiveDriverWGS[25] model both mutational burden and functional consequence but not through

functional impact scores. Three reference driver element sets were used: the COSMIC Cancer Gene Census (CGC)[26,27], the PCAWG raw integrated driver candidates (PCAWG-raw) and the PCAWG consensus driver candidates (PCAWG-consensus). The CGC is a catalogue of driver genes for which mutations have been causally implicated in cancer and was used as the gold standard set (i.e., used in the calculation of precision and recall) for coding and splice site drivers. PCAWG-raw is an integration of driver elements called by 12 different driver detection methods on the same data we used here. PCAWG-consensus is a conservative set derived from the PCAWG-raw by applying multiple stringent filters to control the false discovery rate; in particular, the majority of non-coding candidates from lymphoid tumours and skin melanomas is excluded from this set because of hyper-mutational processes in these tumour types that create prominent mutational hotspots[28–30]. For the same reason our analysis of non-coding regions for tumour-specific and the pan-cancer cohorts excluded melanoma and lymphoma.

Overall, we observed well-calibrated p values in DriverPower's results with or without functional adjustment (Fig. 1d and Supplementary Fig. 10) and a high accuracy for both coding and non-coding driver discovery (Fig. 1e, Supplementary Data 3). For protein-coding regions (CDS), DriverPower found 217 significant ($q < 0.1$) driver candidates. As a gene (e.g., *TP53*) can be driver in multiple cohorts, the unique number of genes was 131. The precision of the algorithm's driver calls was high. Among the driver genes called by DriverPower, 82.5% (179/217) of all genes were present within the CGC. For non-CGC genes, 27 and 3 genes were present within PCAWG-consensus and PCAWG-raw, respectively. Thus, only 3.7% (8/217) coding driver candidates

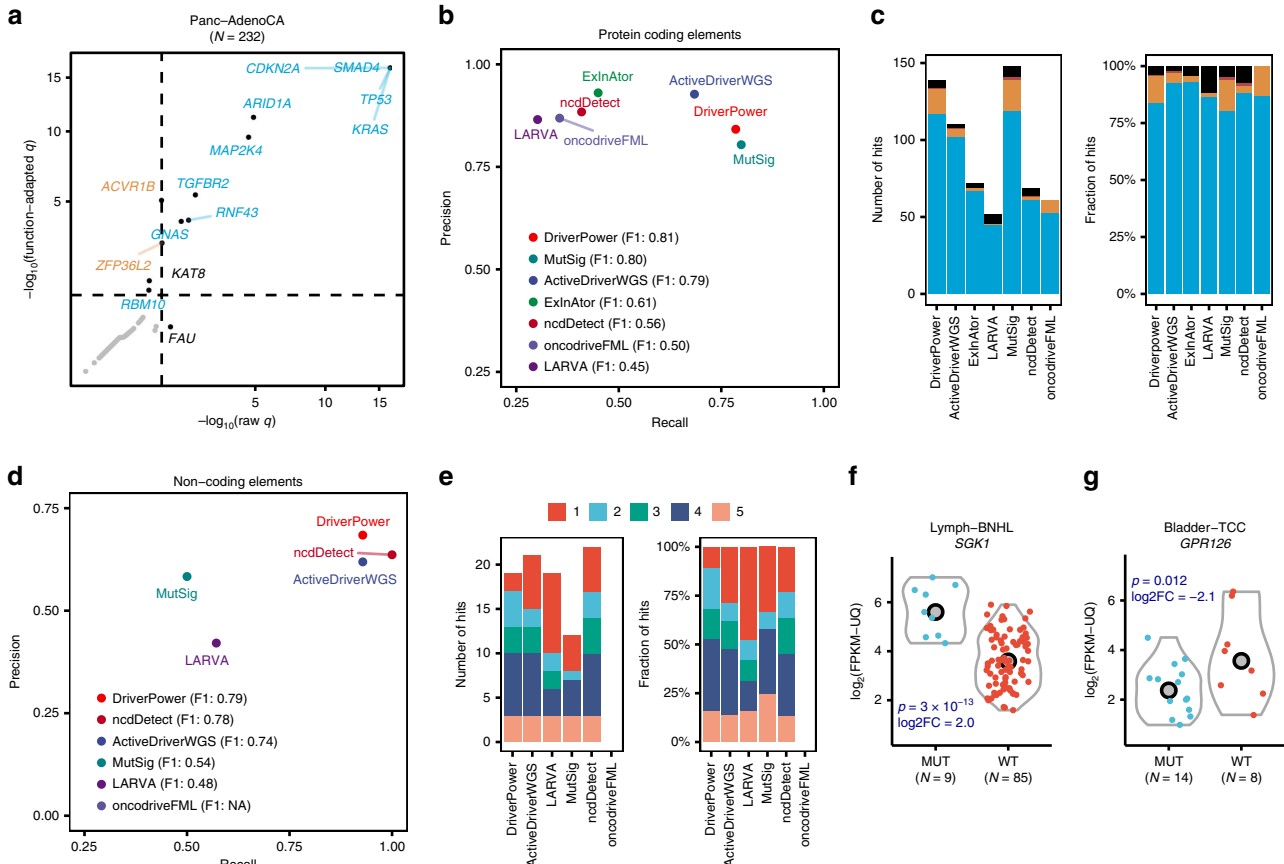

**Fig. 2 Benchmarking DriverPower driver discovery performance. a** Comparison of CDS results with or without functional adjustment for Panc-AdenoCA. Dashed lines in **a** represent the $q$ value = 0.1 threshold. Function-adapted $q$ values are $q$ values with the incorporation of functional impact scores. Only significant genes are labelled (colour legend is the same as Fig. 1e). **b**, **c** Benchmark results for coding genes compared with six other driver discovery methods. **d**, **e** Benchmark results for 3'-UTR, 5'-UTR, promoter and enhancer sets. **b**, **d** Show the precision and recall for each method according to results of 26 tumour cohorts (no melanoma and lymphoma). **c** Shows the number and fraction of coding driver candidates called by each method that are contained within reference gene sets. The coloured columns in **c** correspond to different reference driver sets (colour legend is the same as Fig. 1e). **e** Shows the number and fraction of non-coding driver candidates called by each method that are also called by others. The coloured columns in **e** correspond to the number of methods that agree on a driver candidate. **f** Differential expression analysis for the CDS and splice site of *SGK1* in Lymph-BNHL. **g** Differential expression analysis for the *GPR126* enhancer in Bladder-TCC. MUT indicates samples with mutated element and WT indicates samples without mutated element. Copy number corrected $p$ values from the likelihood ratio test and the log2 fold changes (log2FC) are shown in blue.

called by DriverPower were not contained within any reference gene sets. As expected, incorporation of functional information increased both precision and recall in coding driver discovery (Fig. 2a and Supplementary Fig. 11). For example, in pancreatic ductal adenocarcinoma (Panc-AdenoCA; $N = 232$), the addition of functional adjustment to the algorithm resulted in a gain of three additional drivers (*ACVR1B*, *RBM10* and *ZFP36L2*) and the loss of one likely false-positive genes (*FAU*) (Fig. 2a). Without the use of functional information, the overall precision dropped to 74.6% (156/209) for CGC genes and 88.5% (185/209) for CGC/PCAWG genes. When compared with six other methods using the same 26 non-melanoma/lymphoma cohorts and CGC as the gold standard set, DriverPower (precision = 0.84; recall = 0.79) had the highest F1 score (0.81) (Fig. 2b, c). In our benchmark, sensitivity was a bottleneck for most methods (4/7 with recall < 0.5). When compared with the method with highest recall, the widely used coding driver caller MutSig (precision = 0.80; recall = 0.80), DriverPower identified an additional 21 genes present in CGC (23 for MutSig; Supplementary Fig. 12).

We next benchmarked DriverPower's accuracy for non-coding driver events. For the prediction of driver events affecting the splice sites of coding genes, DriverPower called 47 significant ($q$ <

0.1) candidates with 85.1% (40/47) within CGC. DriverPower (F1 = 0.91) also outperformed two recently published methods, ncdDetect (F1 = 0.65) and oncoDriverFML (F1 = 0.32), for splice site driver detection (Supplementary Fig. 13).

For the prediction of non-coding driver events in 3'-UTRs, 5'-UTRs, promoters and enhancers, DriverPower identified 19 candidates in non-melanoma/lymphoma tumour cohorts and 24 candidates in the pan-cancer cohort. Benchmarking results showed that DriverPower has the highest F1 score (0.79) among the six methods evaluated (Fig. 2d, e). Promoter and 5'-UTR driver candidates called by DriverPower were associated with a total of 17 unique genes. Of these, one gene (*TERT*) was in CGC, four genes (*WDR74*, *HES1*, *MTG2* and *PTDSS1*) were in PCAWG-consensus, and six other genes were in PCAWG-raw. DriverPower also called two 3'-UTR driver candidates in total, including *TOB1* in pan-cancer and *ALB* in Liver-HCC. Both candidates were present in the PCAWG-consensus. For enhancer regions, DriverPower identified two candidates: chr6:142,705,600-142,706,400 (linked to *GPR126*) and chr7:86,865,600-86,866,400 (linked to *TP53TG1*). Both enhancer elements were identified by PCAWG-raw; the *TP53TG1* enhancer was the only enhancer for non-melanoma/lymphoma tumours in the PCAWG-consensus set.

For long non-coding RNA (lncRNA) genes and their promoters, DriverPower found 9 candidates in total. Among them, 6 and 3 were contained within PCAWG-consensus and PCAWG-raw, respectively. These candidates targeted three unique lncRNAs: *RN7SK, RMRP* and *RPPH1*. The promoter of *RMRP* was significantly ($q < 0.1$) mutated in four cohorts (Breast-AdenoCA, Liver-HCC, Stomach-AdenoCA and pan-cancer) and has been nominated as a novel non-coding driver.

**DriverPower-exclusive driver candidates overview**. A total of 11 coding and 17 unique non-coding candidates were exclusively identified by DriverPower (not present in either CGC or PCAWG-consensus; Supplementary Data 4). We sought to evaluate these exclusive driver candidates using literature evidence and correlative orthogonal data such as the effect of the variant on RNA-seq expression levels and the presence of somatic copy number alterations (SCNAs) and somatic structural variations (SVs) covering the same regions. On this basis, we found that many of the DriverPower-exclusive candidates are plausible cancer drivers.

Among protein-coding genes, DriverPower identified *EEF1A2* (eukaryotic translation elongation factor 1 alpha 2) in the oesophageal adenocarcinoma cohort (Eso-AdenoCA; 7/95 samples). All seven observed mutations were missense (Supplementary Fig. 14a). Although no RNA-seq data are available for Eso-AdenoCA samples, SCNA analysis indicated that *EEF1A2* is amplified in 69.5% (66/95) of Eso-AdenoCA samples (vs. 27.9% of non-Eso-AdenoCA samples; Supplementary Fig. 14b), suggesting a potential gain-of-function role in this cancer type. The amplification of *EEF1A2* (20q13.33) was also confirmed by the GISTIC2.0 ($q = 0.0006$). The same 1-Mbp locus detected by GISTIC2.0 was also amplified recurrently in other tumour types, including 73.1% of colorectal adenocarcinoma, 64.7% of stomach adenocarcinoma and 55.4% of ovarian adenocarcinoma. Supporting this hypothesis, previous studies have also demonstrated that *EEF1A2* is a putative oncogene in ovarian cancer and overexpressed in various tumour types[31–33].

Another protein-coding gene exclusively identified by DriverPower was *MEF2B* in B-cell non-Hodgkin's leukaemia (Lymph-BNHL; 8/105 samples). Among nine observed mutations, eight mutations were missense and one was a frameshift deletion (Supplementary Fig. 14c). RNA-seq data confirmed that mutated samples overexpressed *MEF2B* (copy number corrected $p = 0.011$; Supplementary Fig. 14d). *MEF2B* (Myocyte enhancer factor 2B) has been identified in multiple WES studies[34–36], and a previous study has also shown that *MEF2B* mutations can dysregulate cell migration in non-Hodgkin lymphoma[37].

One splice site candidate exclusively called by DriverPower is *SGK1* (serum/glucocorticoid regulated kinase 1) in Lymph-BNHL. The same gene was also significant in DriverPower's CDS result for Lymph-BNHL (Supplementary Fig. 14e), resulting in a total of 13.3% (14/105) Lymph-BNHL samples being affected by non-synonymous or splice site mutations in *SGK1*. *SGK1* is present in PCAWG-raw but was filtered out owing to the large number of AID-related variants in this tumour cohort. However, differential expression analysis indicated that *SGK1* is significantly overexpressed in mutated Lymph-BNHL samples relative to non-mutated samples (copy number corrected $p = 3e$-13 from likelihood ratio test; Fig. 2f). *SGK1* encodes a serine/threonine protein kinase that has an important role in cellular stress response and its CDS has been nominated as a driver in earlier WES studies[35,36]. Another study has also demonstrated that the administration of an *SGK1* inhibitor induces apoptosis in lymphoma cell lines[38]. Together these data support a potential driver role for *SGK1* in Lymph-BNHL.

The *GPR126* (adhesion G protein-coupled receptor G6) enhancer candidate was filtered out from the PCAWG-raw set because of mutations in palindrome loops, which makes it unclear whether mutations in the *GPR126* enhancer are caused by mutational mechanism associated with palindrome loops or positive selection. We found that the *GPR126* enhancer is recurrently mutated in transitional cell carcinoma of the bladder (Bladder-TCC; 14/23 samples) and breast adenocarcinoma (Breast-AdenoCA; 8/195) (Supplementary Fig. 14f). *GPR126* is among the MammaPrint 70 gene panel used to predict the risk of breast cancer metastasis[39,40]. A study also shows that knockdown of *GPR126* can inhibit the hypoxia-induced angiogenesis in model organisms[41]. Differential expression analysis demonstrated that the *GPR126* is significantly downregulated in Bladder-TCC samples with enhancer mutations (copy number corrected $p = 0.012$ from likelihood ratio test; Fig. 2g) relative to those carrying the wild-type enhancer, suggesting a functional role for these mutations.

Several somatically altered histone genes have been implicated in human cancer, such as *H3F3A* (identified as a pan-cancer driver in this study), *H3F3B* and *HIST1H3B*[42–44]. DriverPower identified four histone genes as driver candidates in the pan-cancer cohort, two of which were absent from CGC or PCAWG-consensus: the 5′-UTR of *HIST1H2AC* and *HIST1H2BD*. Previous studies have shown that the protein levels of the replication-dependent histone H2A variant *HIST1H2AC* (encoding histone H2A type 1-C) is decreased in chronic lymphocytic leukaemia patients and bladder cancer cells[45,46], and the siRNA knockdown of *HIST1H2AC* increases cell proliferation and promote oncogenesis[46].

Several other driver candidates exclusively called by DriverPower are associated with genes that may have a role in cancer. The highly expressed liver-specific gene *ALB* (albumin) is significant (DriverPower $q < 0.1$) for somatic mutations affecting its CDS, splice site, 3′-UTR and promoter in Liver-HCC; the splice site and promoter (under CADD scores) were discovered by DriverPower exclusively. Correlative evidence from gene expression and copy number alterations suggested that loss-of-function mutations in *ALB* are subject to positive selection in Liver-HCC as described elsewhere[47]. The CDS of *KAT8* (lysine acetyltransferase 8) was called by DriverPower in Panc-AdenoCA with 100% (5/5) missense mutations. As a histone modifier, *KAT8* has been shown to physically interact with *MLL1* and regulate known cancer drivers *ATM* and *TP53*[48–51]. Previous studies have also shown that *KAT8* is downregulated in gastric cancer[52] and *KAT8* can suppress tumour progression by inhibiting epithelial-to-mesenchymal transition[53]. The 5′-UTR and promoter of *SRSF9* (serine and arginine rich splicing factor 9) was significant in DriverPower's results for pan-cancer and not present in any reference driver sets. The protein encoded by *SRSF9* is part of the spliceosome; a previous study indicates that the proto-oncogene *SRSF9* is overexpressed in multiple tumours and that this overexpression can cause the accumulation of β-Catenin[54]. The same study also showed that the depletion of *SRSF9* proteins could inhibit colon cancer cell proliferation.

In summary, 4/11 coding and 4/17 unique non-coding driver candidates exclusively called by DriverPower had some form of support from the literature or orthogonal evidence. If we assume that all the exclusive candidates that lack such evidence are false positives, then this puts an estimate of DriverPower's false discovery rate across the PCAWG data set at 3.2% (7/217) for coding and 16.8% (16/95) for non-coding regions. However, this assumption is probably invalid as most of these lack-of-evidence candidates are also identified by other methods and present in PCAWG-raw. We acknowledge that lack-of-evidence candidates may contain false-positive calls, but they may also contain previously unknown drivers. For example, the 5′-UTR of

*TBC1D12* in Breast-AdenoCA, which has been filtered out from the PCAWG-raw owing to possible hypermutability, is called by all but one driver discovery methods and is reported as a putative cancer driver in previous studies because of two recurrent mutations in the Kozak consensus sequence involving in the initiation of protein translation[23,55]. Moreover, according to another recent study, the same *TBC1D12* candidate is still statistically significant in breast cancer even after removing hypermutations, but whether these mutations can alter protein translation in cancer is still undetermined[24]. Some lack-of-evidence candidates may also fit the mini-driver model of cancer evolution[56]. Unlike classical drivers, mini-drivers can only weakly promote and are not essential for tumour progression, hence present at a lower frequency in cancer cohorts. Further investigation is required to determine the role of lack-of-evidence candidates in cancer.

**DriverPower applied to WGS.** To demonstrate the robustness of DriverPower, we applied DriverPower to two public whole-exome sequencing (WES) data sets (Supplementary Fig. 15). Both WES data sets are processed differently than the PCAWG data and contain samples not included in the PCAWG study. For liver cancer, using models trained for Liver-HCC ($N = 314$), DriverPower identified 14 coding drivers from 364 TCGA-LIHC samples (53 shared with Liver-HCC). All but one driver candidates were present within the CGC or PCAWG-consensus. For pancreatic adenocarcinoma, using models trained for Panc-AdenoCA ($N = 232$), DriverPower identified six coding drivers from 180 TCGA-PAAD samples (no shared samples with the PCAWG study) and all corresponded to known driver genes.

## Discussion

Computational driver discovery is essential to distinguish driver from passenger mutations in the coding and non-coding regions of whole cancer genomes. Here we report DriverPower, a new framework for accurately identifying both types of driver mutation by combining mutational burden and functional impact information. The method takes advantage of the large somatic mutation sets produced by WGS technology to build an accurate global BMR model from more than a thousand genomic features. This contrasts with methods that build a local BMR model using selected or flanking regions. One advantage of this is that the method is not biased towards coding regions, but uses the same model for coding and non-coding cancer driver discovery. Another advantage is the method's high degree of modularity. DriverPower can potentially work with any types of genomic element (contiguous or disjoint, coding or non-coding, proximate or distal to genes), any regression algorithms for modelling BMR and any functional impact score scheme. Although DriverPower is designed for WGS projects, it performs robustly in WES strategies as well.

In comparison with the other driver discovery methods evaluated by the PCAWG Drivers and Functional Interpretation Working Group, DriverPower provides the best balance of precision and recall, although is not always the top-ranked method when either metric is considered independently (Fig. 2b, d). As discussed in Supplementary Note 1, DriverPower is parameterised to allow for adjustment of the precision-recall trade-off; in this study, we selected conservative parameters that prioritise precision over recall especially for non-coding regions (Supplementary Fig. 16).

There are several ways in which the accuracy of DriverPower could be improved. One approach to improve recall is to take into the account the potential presence of negative (purifying) selection in the functional regions selected for testing. When the BMR model is trained, we use random genomic elements that are predominantly under neutral selection. However, the functional elements selected for testing are more likely to be under positive and/or negative selection[57]. The observed mutation rate reflects the balance between positive and negative selection, and negative selection at one site in the element will diminish the signal of positive selection at other sites, reducing the sensitivity of the method as a whole. To our knowledge, no driver discovery tool currently models the effect of negatively selected sites; future work aims to take this mechanism into account.

The precision of the method can also be improved. False-positive driver calls may be caused by technical errors such as variant-calling artefacts that artificially increase the local mutation rate, or by biological processes that are not captured by the BMR model such as regional differences in activation-induced cytidine deaminase (AID) activity. These can potentially be mitigated by incorporating into the BMR model additional features relevant to the technical and biological processes. For example, incorporating read-level coverage, mapping and bias scores into the BMR could help correct for regions prone to variant-calling artefacts, whereas features like the number of palindrome loops and the fraction of mutations caused by AID per element could be used to adjust for locally-acting hypermutation processes.

When applied to the PCAWG data set, DriverPower called nearly twice as many coding driver events as non-coding ones, a ratio also observed by the PCAWG driver study, a ratio also observed by the PCAWG driver study[47]. Although this unbalanced ratio may reflect cancer biology, there is also the possibility that it reflects, at least in part, the technical challenge of sequencing and interpreting non-coding regions. Potential artefacts include systematic undercalling of somatic variants in non-coding regions[24], a problem that could be rectified by deeper coverage. For example, it is estimated that ~ 216 mutations in the *TERT* promoter are likely to be missed in the PCAWG data set owing to low detection sensitivity[47]. Another technical issue is raised by the fact that several non-coding candidates are only significant in the pan-cancer cohort, suggesting that the data set is statistically underpowered. In fact, although we studied 2583 genomes here, many tumour types have a sample size fewer than 30. To overcome this issue, we could either sequence more genomes or reduce the size of the set of test elements by narrowing it to functional motifs or conserved bases[58]. Moreover, only ~ 3.7% of the genome has been studied here. There may be more non-coding drivers in other types of regulatory elements, which demands more complete annotations for the non-coding part of the human genome. At last, functional impact score schemes are currently biased toward coding mutations; therefore, improved functional scoring schemes will also help us identify more functionally relevant non-coding cancer drivers in the future.

A comprehensive catalogue of coding and non-coding cancer drivers will accelerate the clinical translation of cancer genomic study to precision medicine. As more cancer genomes and more cancer types are sequenced, a general and accurate framework for computational driver discovery like DriverPower will become increasingly useful.

## Methods

**Ethical review.** Sequencing of human subjects tissue was performed by ICGC and TCGA consortium members under a series of locally approved Institutional Review Board (IRB) protocols as described in Hudson et al.[59]. Informed consent was obtained from all human participants. Ethical review of the current data analysis project was granted by the University of Toronto Research Ethics Board (REB) under protocol #30278, 'Pan-cancer Analysis of Whole Genomes: PCAWG'.

**Generation of cancer whole-genome somatic mutations.** All DNA somatic SNVs and indels for 2583 donors were obtained from the PCAWG project (somatic variant callset released October 2016)[9]. For our analysis, donors with hypermutated signatures were excluded ($n = 69$, defined as > 30 mutations per Mb). Otherwise,

we used the same type-specific ($n = 29$) and pan-cancer (all tumour samples except Skin-Melanoma, Lymph-NOS, Lymph-CLL and Lymph-BNHL) sample cohorts as the PCAWG Drivers and Functional Interpretation Working Group (PDFIWG; Supplementary Data 1)[47].

**Generation of simulated somatic mutations**. We used three simulated data sets (Broad, DKFZ and Sanger simulations) from the PDFIWG (described in detail at Rheinbay et al.[47]). These simulations were made to capture the variation of BMR and remove the signal of positive selection through permutations of observed somatic mutations.

**Generation of test and training genomic elements**. We define a genomic element as the collection of genome coordinates that defines one specific functional region of interest. For example, the CDS element of *TP53* is the combination of all protein-coding regions in *TP53*.

We used eight test element sets in our analysis, including the CDS ($n = 20,185$), splice site ($n = 18,729$), 5′-UTR ($n = 19,369$), 3′-UTR ($n = 19,188$), promoter ($n = 20,164$), enhancer ($n = 30,816$), lncRNA ($n = 5,580$) and lncRNA promoter ($n = 5373$). All test element sets were obtained from the PCAWG project. GENCODE v19 was used as the reference gene model when building those sets[60]. Non-coding RNA annotations were collected from multiple sources as described.

We constructed genomic element training sets by randomly sampling genome coordinates from hg19, the build used for PCAWG. The length of each training element was sampled from the length distribution of test elements and multiplied by a factor of 3. Training elements overlapping test elements were removed. In total, 867,266 training elements were created and ~ 54% (1,545,491,997 bp) of the genome was covered.

**Collection and generation of features**. We collected 1373 features in total. Details including data sources can be found at Supplementary Data 2. Nucleotide content features were calculated as the fraction of 2-mers and 3-mers in each genomic element. The number of 2-mers and 3-mers was counted directly from genome sequences (hg19). For raw features in bigwig format (typically genome-wide signals), we calculated the average signal strength of covered bases in each element using the bigWigAverageOverBed (v2) utility from the UCSC genome browser[61]. For raw features in BED format (typically narrow peaks of ChIP-seq data), we calculated the percentage of bases intersecting BED for each element with the BEDTools (v2.24.0)[62]. All missing values in features were filled with 0.

**The DriverPower outline**. The main steps of the DriverPower (v1.0.0) framework are summarised below. Details of each step are described in following sections. The difference between v1.0.0 and the version used in the PDFIWG data analysis freeze (April 2017) is discussed in Supplementary Note 2 and Supplementary Fig. 17.

The first step of DriverPower is to scale features and/or filter out excluded regions. The second step is to build the BMR model using the GBM, or randomised lasso followed by binomial GLMs. The purpose of the BMR model is to estimate the expected number of mutations ($y^\wedge$) for any genomic element. Namely, we want to obtain $y_i^\wedge = E(y_i|X_i, L_i)$ where $X_i$ and $L_i$ are the feature vector and length for the element $i$. The third step is to conduct burden test with observed ($y$) and predicted ($y^\wedge$) mutation counts, and perform multiple testing correction. The fourth step is to adjust observed mutation counts ($y$) based on functional impact scores for nearly significant elements ($q < 0.25$). The last step is to re-assess the significance for nearly significant elements with functional adjusted mutation counts followed by multiple testing correction.

**Scaling of features**. Features were scaled with RobustScaler from scikit-learn (version 0.18)[63]. Feature scaling was only conducted for randomised lasso and GLMs.

**Definition of excluded regions**. In this study, all bases in the excluded regions were removed before any analysis. The excluded region consists of three sets: (1) all $N$ bases and gaps in the hg19 genome (fetched from the UCSC table browser[12]); (2) ENCODE consensus excludable regions (the DAC Blacklisted Regions track and the Duke Excluded Regions track from the UCSC genome browser)[64]; (3) PCAWG low mappability regions (data retrieved from the PCAWG variant group). PCAWG low mappability regions are defined as regions callable in fewer than 556/1111 (~ 50%) tumour–normal pairs. For each tumour–normal pair, a base is callable if there are more than 14/8 high quality reads in tumour/normal WGS. In total, 2,806,377,226 bp, or 96.41% of the genome are defined as callable.

**Feature selection with randomised lasso**. To select features, we randomly sub-sampled 10% of the training set 500 times. Then for the $k$-th subset with size $N_k$, the following model was fitted[65]:

$$w_k^\wedge = arg\,\underset{w}{min}\left(\frac{1}{N_k}\left\|logit\left(\frac{y + 1/2}{N \cdot L}\right) - Xw\right\|_2^2 + \alpha \sum_{i=1}^{1373}\frac{|w_i|}{b_i}\right)$$

where $N$ is the total number of donors in the data set, $X$ is the feature matrix, $w$ is

the weight vector, $\alpha$ is the regularisation parameter, and $b_i$ is the scaling factor. The regularisation parameter $\alpha$ was determined by a fivefold CV lasso with 33% of the training data. For the $k$-th sub-sampling, the ith feature was selected if $w_{ki}^\wedge \geq 0.001$. The final feature importance score was calculated as the fraction of times that a feature was selected. Only features with score $> 0.5$ were used in the GLM BMR model.

**Prediction of the BMR with GLM**. When using the GLM, we modelled the observed mutations in each genomic element with a binomial distribution, that is

$$y \sim B(n, p)\ with\ n = N \cdot L\ and\ p = y^\wedge/(N \cdot L)$$

where $y$ is the observed mutation count and $y^\wedge$ is the estimated mutation count. We used the binomial GLM to obtain $y^\wedge$ with the logit link function, that is

$$\frac{\hat{y}}{N \cdot L} = E\left(\frac{y}{N \cdot L}\Big|X^{select}\right) = logit^{-1}(X^{select}\beta)$$

where $X^{select}$ is the selected feature matrix and $\beta$ is the regression coefficient vector.

**Prediction of the BMR with GBM**. We trained the GBM with XGBoost[66]. All features were used in model training. The negative Poisson log-likelihood was chosen as the objective function and $ln(N \cdot L)$ of elements were used as offset (i.e., base_margin in XGBoost). Other non-default parameters used in DriverPower were as follows: eta = 0.05, max_depth = 8, subsample = 0.6, max_delta_step = 1.2, early_stop_rounds = 5 and nrounds = 5000. The feature importance for GBM is measured by the improvement in accuracy brought by a feature across all trees. XGBoost returns feature importance that sums up to 1 for all features. We also normalised the importance to a [0, 1] scale (i.e., importance relative to the most important feature).

**Evaluation of two BMR models**. We evaluated both models with 1 Mb autosome bins ($n = 2521$) and training genomic elements ($n = 867,266$) defined above. The 1 Mb elements have been used in many studies[14,15,67]. For both elements, we obtain the predicted mutation rate by fivefold CV. For 1 Mb elements, we used fourfold data for model training and onefold data for model evaluation. For training elements, we use 1-fold data to train the model and the rest to evaluate. As per previous work, we used $R^2$ score and Pearson's $r$ as evaluation metrics[15]. Standard error of the mean (SEM) for $R^2$ and $r$ was calculated from fivefold CV.

**Calculation of element functional impact scores**. Four different functional scores were used in this analysis[16–19]. For CDS, CADD (SNVs and indels, v1.3), DANN (SNVs) and EIGEN (SNVs) scores were used. CADD indel scores were generated with the CADD web interface for all observed indels in the PCAWG data set. For splice site, CADD and DANN scores were used. For non-coding elements, the CADD, DANN and LINSIGHT (SNVs and indels) score were used. Then the following steps were used to calculate the functional impact score per genomic element. First, raw scores were retrieved for all observed mutations in the data set. Second, all raw scores were converted to phred-like scores by $-10log_{10}(rank/N_m)$, where $N_m$ is the number of observed mutations having scores. Third, for each genomic element, its functional score $S$ was calculated as:

$$S = \frac{1}{N}\sum_{i=1}^{N}s_i$$

where $N$ is the number of donors and $s_i$ is the average functional impact score for the $i$th donor.

**Adjustment of the mutation count**. To compensate for the unbalanced number of mutations among samples, instead of using the mutation count per element directly we used the geometric mean of mutation count and sample count. That is, we use the balanced count $y^b = \sqrt{y \cdot n_d}$ instead of $y$ directly for significance test, where $n_d$ is the number of mutated donors. Based on the motivation that not all mutations should be weighted the same, the balanced mutation count $y^b$ was then adjusted for nearly significant elements (raw $q$ value $< 0.25$) by a functional weight $w$, that is $y^f = w \cdot y^b$, where $y^f$ is the functionally adjusted mutation count. For the element $j$, the functional weight $w_j$ was calculated based on its functional score $S_j$ and a threshold score $S_T$:

$$w_j = \frac{S_j}{S_T} = \frac{S_j}{-10log_{10}F}$$

The threshold score $S_T$ is controlled by a single parameter $F$ between 0 and 1, and can be interpreted as the fraction of functionally relevant variants among all observed variants. Parameter tuning of $F$ can be found at Supplementary Note 1.

**Assessment of the element significance**. For each element, we calculated $P(y^b \geq y^\wedge)$ as the raw $p$ value and $P(y^f \geq y^\wedge)$ as the function-adapted $p$ value. As over-dispersion has been documented in burden-based methods and can affect the driver discovery accuracy[22], here we performed a regression-based overdispersion test for each tumour cohort using the training set[68]. Based on the result of the

overdispersion test, we calculated the raw and function-adapted $p$ values by following a binomial distribution or a negative binomial distribution:

$$y^b \, or \, y^f \sim \begin{cases} NB(y^\wedge, s \cdot \theta), if \, p \le 0.01 \\ B\left(N \cdot L, \frac{y^\wedge}{N \cdot L}\right) \end{cases}$$

otherwise, where $p$ and $\theta$ are the $p$ value and dispersion parameter estimated from the overdispersion test, and $s$ is the scaling factor for $\theta$ used to accommodate the discrepancy between test and training set in terms of the dispersion level. We used $s = 3$ for lymphomas and $s = 1$ otherwise in this analysis.

**Multiple testing correction**. In all cases, $q$ values were generated by the Benjamini–Hochberg procedure[69]. We chose $q < 0.1$ as the significant level and $q < 0.25$ as the nearly significant level. For each element set, multiple testing correction was performed for each tumour cohort (cohort $q$ value) and across all tumour cohorts (global $q$ value). Cohort $q$ values were used in functional adjustment and global $q$ values were used to define the final driver list.

**Generation of reference cancer drivers**. Reference cancer drivers were used to benchmark the performance of DriverPower. Three reference sets were used: (1) the COSMIC CGC (v82, $n = 567$); (2) the PCAWG consensus driver candidates (PCAWG-consensus; $n = 157$ for coding and $n = 26$ for non-coding); (3) the PCAWG raw integrated driver candidates (PCAWG-raw; $n = 193$ for coding and $n = 79$ for non-coding). PCAWG-consensus ($q$ value post-filtering < 0.1) is a set of highly confident non-coding drivers and subjected to multiple stringent filters as described. PCAWG-raw ($q$ value pre-filtering < 0.1) is a superset of PCAWG-consensus and includes non-coding drivers that were not subjected to the filtering process. PCAWG-raw driver candidates that are mutated in fewer than three samples were removed in this analysis. For promoter and 5′-UTR candidates in the PCAWG consensus drivers, we reversed the filtering for overlapping elements (i.e., one element is selected over the overlapping element based on prior knowledge). For example, we kept both the promoter and the overlapping 5′-UTR of *WDR74* in this analysis; in the PCAWG consensus set, the *WDR74* promoter is preferentially selected over its 5′-UTR.

**Benchmarking of DriverPower**. We compared coding and non-coding driver candidates called by DriverPower to driver candidates called by six other published driver detection tools (ActiveDriverWGS[25], ExInAtor[20], LARVA[22], MutSig[24], ncdDetect[21] and oncodriveFML[23]). Driver calls for 26 single tumour cohorts (no Skin-Melanoma, Lymph-CLL and Lymph-BNHL) were retrieved from the PCAWG driver group. For each method, we removed driver candidates that are mutated in fewer than three samples. We used precision (TP/(TP + FP)), recall (TP/(TP + FN)) and F1 score (2*Precision*Recall/(Precision + Recall)) as performance metrics.

For CDS, we used the CGC gene set as the gold standard. For each method, true positive genes were defined as genes presented in the gold standard set and the precision was then calculated as the fraction of true positive genes among all called genes. For recall, since we cannot accurately know the expected set of driver genes that should be called for each tumour cohort in the data set, a lower-bound approximation was used instead. The lower-bound approximation was estimated by taking the union of all true positive genes identified by each method and the recall was then calculated as the fraction of true positive genes called among the lower-bound approximation.

For gene splice sites, the same gold standard gene set and benchmark method as CDS were used. Owing to data availability, the comparison was only performed for ncdDetect, oncodriveFML and DriverPower.

For promoters, enhancers, 3′-UTRs and 5′-UTRs, because the number of non-coding driver candidates is small, four element sets were benchmarked together. No data for ExInAtor is available for this comparison. For each tumour cohort, true positive driver elements were defined as elements called by at least three methods. The calculation of precision, recall and F1 score was then identical as for the CDS and splice site.

**Somatic copy number and SVs analysis**. We used SCNA (including GISTIC2.0 results) and SV call sets released January 2017[70]. The copy number status (loss, neutral or gain) of a region is classified based on the difference between the absolute copy number of the region and the genome-wide ploidy of the donor. For gene-level SVs, we calculated the number of breakpoints per gene (including CDS, splice sites, UTRs and promoters) per donor.

**Differential expression analysis**. We used the upper quartile normalised gene expression (FPKM-UQ) released May 2016[71]. When comparing the expression difference between two groups of donors, we fitted the following quasi-Poisson family GLM and then employed the likelihood ratio test to obtain $p$ values for mutational status: FPKM-UQ ~ MUT + SCNA + [Tissue], where MUT is the mutational status (0 for unmutated donors and 1 for mutated donors), SCNA is the somatic copy number status ($-1$, 0 or 1 for copy number loss, neutral or gain, respectively) and Tissue is the tumour tissue type. The tissue type was only used for pan-cancer comparison for the adjustment of tumour types.

**WES data analysis**. We obtained two WES data sets through the Genomic Data Common (GDC)[72]: TCGA-PAAD (35,321 somatic mutations across 180 samples) and TCGA-LIHA (56,208 somatic mutations across 364 samples). We chose public MuTect2 variants from GDC. Variant coordinates were lifted from hg38 to hg19 with the UCSC liftOver tool. Only CADD scores were used to detect drivers. For TCGA-PAAD, GBM models trained from Panc-AdenoCA of the PCAWG data were used. For TCGA-LIHA, GBM models trained from Liver-HCC were used.

**Reporting summary**. Further information on research design is available in the Nature Research Reporting Summary linked to this article.

## Code availability

The source code for DriverPower (written mainly in Python 3) is available at GitHub [https://github.com/smshuai/DriverPower]. It is distributed under GNU General Public License 3.0, which allows for reuse and redistribution. Other software packages and bioinformatics tools used in this study are indicated in the corresponding method sections. The core computational pipelines used by the PCAWG Consortium for alignment, quality control and variant calling are available to the public at [https://dockstore.org/search?search=pcawg] under the GNU General Public License v3.0, which allows for reuse and distribution.

## Data availability

The data sets underpinning the analyses in the paper are detailed in Supplementary Table 1. Aligned sequencing data, as well as somatic and germline variant calls from PCAWG tumours, including SNVs, indels, copy number alterations and structural variants, are available for download at [https://dcc.icgc.org/releases/PCAWG]. Additional information on accessing the data, including raw read files, can be found at [https://docs.icgc.org/pcawg/data/]. In accordance with the data access policies of the ICGC and TCGA projects, most molecular, clinical and specimen data are in an open tier, which does not require access approval. To access potentially identification information, such as germline alleles and underlying sequencing data, researchers will need to apply to the TCGA Data Access Committee (DAC) via dbGaP [https://dbgap.ncbi.nlm.nih.gov/aa/wga.cgi?page=login] for access to the TCGA portion of the data set, and to the ICGC Data Access Compliance Office (DACO; [https://icgc.org/daco]) for the ICGC portion. In addition, to access somatic SNVs derived from TCGA donors, researchers will also need to obtain dbGaP authorisation.

In addition, the analyses in this paper used several data sets that were derived from the raw sequencing data and variant calls (Supplementary Table 1). The individual data sets are available at Synapse (https://www.synapse.org/), and are denoted with synXXXXX accession numbers (listed under Synapse ID); all these data sets are also mirrored at https://dcc.icgc.org, with full links, filenames, accession numbers and descriptions detailed in Supplementary Table 1. The data sets encompass: harmonised tumour histopathology annotations using a standardised hierarchical ontology (syn10389164); consensus somatic SNVs, MNVs and indels (syn7364923); gene expression profiles from RNA-sequencing data (syn5553991); genomic intervals used in driver region calls (syn5259890); driver region calls by each individual methods (syn7359546); consensus gene-level somatic copy number calls (syn8239175); three simulation data sets for somatic mutations (syn7187923, syn7436065 and syn7152699).

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

## Acknowledgements

This work was supported in part by the Government of Ontario. We acknowledge the contributions of the many clinical networks across ICGC and TCGA who provided samples and data to the PCAWG Consortium, and the contributions of the Technical Working Group and the Germline Working Group of the PCAWG Consortium for collation, realignment and harmonised variant calling of the cancer genomes used in this study. We thank the patients and their families for their participation in the individual ICGC and TCGA projects.

## Author contributions

S.S. and L.S. conceived the study and wrote the paper. S.S. designed the software and conducted the data analysis. L.S. and S.G. supervised the project. Input for driver region calls and driver candidates called by other methods were made available by the PCAWG Drivers and Functional Interpretation Working Group, where Mark Gerstein, Gad Getz, Michael S. Lawrence, Jakob Skou Pedersen, Benjamin J. Raphael, Joshua M. Stuart and David A. Wheeler were working group co-leaders.

## Competing interests

The authors declare no competing interests.

## Additional information

## PCAWG Drivers and Functional Interpretation Working Group

Federico Abascal[5], Samirkumar B. Amin[6,7,8], Gary D. Bader[1], Pratiti Bandopadhayay[9,10,11], Jonathan Barenboim[2], Rameen Beroukhim[9,12,13], Johanna Bertl[14,15], Keith A. Boroevich[16,17], Søren Brunak[18,19], Peter J. Campbell[5,20], Joana Carlevaro-Fita[21,22,23], Dimple Chakravarty[24,25], Calvin Wing Yiu Chan[26,27], Ken Chen[28], Jung Kyoon Choi[29], Jordi Deu-Pons[30,31], Priyanka Dhingra[32,33], Klev Diamanti[34], Lars Feuerbach[35], J. Lynn Fink[36,37], Nuno A. Fonseca[38,39], Joan Frigola[30], Carlo Gambacorti-Passerini[40], Dale W. Garsed[41,42], Mark Gerstein[43,44,45], Gad Getz[9,13,46,47], Qianyun Guo[48], Ivo G. Gut[49,50], David Haan[51], Mark P. Hamilton[52], Nicholas J. Haradhvala[9,53], Arif O. Harmanci[45,54], Mohamed Helmy[55], Carl Herrmann[26,56,57], Julian M. Hess[9,58], Asger Hobolth[14,48], Ermin Hodzic[59], Chen Hong[27,35], Henrik Hornshøj[15], Keren Isaev[2,60], Jose M.G. Izarzugaza[18], Rory Johnson[21,22], Todd A. Johnson[16], Malene Juul[15], Randi Istrup Juul[15], Andre Kahles[61,62,63,64,65], Abdullah Kahraman[66,67,68], Manolis Kellis[9,69], Ekta Khurana[32,33,70,71], Jaegil Kim[9], Jong K. Kim[72], Youngwook Kim[73,74], Jan Komorowski[34,75], Jan O. Korbel[39,76], Sushant Kumar[44,45], Andrés Lanzós[21,22,23], Erik Larsson[77], Michael S. Lawrence[9,16,53], Donghoon Lee[45], Kjong-Van Lehmann[61,63,64,65,78], Shantao Li[45], Xiaotong Li[45], Ziao Lin[9,79], Eric Minwei Liu[32,33,80], Lucas Lochovsky[8,44,45], Shaoke Lou[44,45], Tobias Madsen[15], Kathleen Marchal[81,82], Iñigo Martincorena[5], Alexander Martinez-Fundichely[32,33,71], Yosef E. Maruvka[9,53,58], Patrick D. McGillivray[44], William Meyerson[45,83], Ferran Muiños[31,84], Loris Mularoni[31,84], Hidewaki Nakagawa[17], Morten Muhlig Nielsen[15], Marta Paczkowska[2], Keunchil Park[85,86], Kiejung Park[87], Jakob Skou Pedersen[15,48], Tirso Pons[88], Sergio Pulido-Tamayo[81,82], Benjamin J. Raphael[89], Jüri Reimand[2,60], Iker Reyes-Salazar[84], Matthew A. Reyna[89], Esther Rheinbay[9,13,53], Mark A. Rubin[21,90,91,92,93], Carlota Rubio-Perez[31,84,94], S. Cenk Sahinalp[59,95,96], Gordon Saksena[9], Leonidas Salichos[44,45], Chris Sander[61,97,98], Steven E. Schumacher[9,99], Mark Shackleton[41,42], Ofer Shapira[9,99], Ciyue Shen[98,100], Raunak Shrestha[96], Shimin Shuai[1,2], Nikos Sidiropoulos[101], Lina Sieverling[27,35], Nasa Sinnott-Armstrong[9,102], Lincoln D. Stein[1,2], Joshua M. Stuart[51], David Tamborero[31,84], Grace Tiao[9], Tatsuhiko Tsunoda[16,103,104,105], Husen M. Umer[34,106],

Liis Uusküla-Reimand[107,108], Alfonso Valencia[36,109], Miguel Vazquez[36,110], Lieven P.C. Verbeke[82,111], Claes Wadelius[112], Lina Wadi[2], Jiayin Wang[113,114,115], Jonathan Warrell[44,45], Sebastian M. Waszak[76], Joachim Weischenfeldt[76,101,116], David A. Wheeler[117,118], Guanming Wu[119], Jun Yu[120], Jing Zhang[45], Xuanping Zhang[113,121], Yan Zhang[45,122,123], Zhongming Zhao[124], Lihua Zou[125] & Christian von Mering[68,126]

[5]Wellcome Sanger Institute, Wellcome Genome Campus, Cambridge CB10 1SA, UK. [6]Department of Genomic Medicine, The University of Texas MD Anderson Cancer Center, Houston, TX 77030, USA. [7]Quantitative & Computational Biosciences Graduate Program, Baylor College of Medicine, Houston, TX 77030, USA. [8]The Jackson Laboratory for Genomic Medicine, Farmington, CT 06032, USA. [9]Broad Institute of MIT and Harvard, Cambridge, MA 02142, USA. [10]Dana-Farber/Boston Children's Cancer and Blood Disorders Center, Boston, MA 02215, USA. [11]Department of Pediatrics, Harvard Medical School, Boston, MA 02115, USA. [12]Department of Medical Oncology, Dana-Farber Cancer Institute, Boston, MA 02115, USA. [13]Harvard Medical School, Boston, MA 02115, USA. [14]Department of Mathematics, Aarhus University, Aarhus 8000, Denmark. [15]Department of Molecular Medicine (MOMA), Aarhus University Hospital, Aarhus N 8200, Denmark. [16]Laboratory for Medical Science Mathematics, RIKEN Center for Integrative Medical Sciences, Yokohama, Kanagawa 230-0045, Japan. [17]RIKEN Center for Integrative Medical Sciences, Yokohama, Kanagawa 230-0045, Japan. [18]Technical University of Denmark, Lyngby 2800, Denmark. [19]University of Copenhagen, Copenhagen 2200, Denmark. [20]Department of Haematology, University of Cambridge, Cambridge CB2 2XY, UK. [21]Department for BioMedical Research, University of Bern, Bern 3008, Switzerland. [22]Department of Medical Oncology, Inselspital, University Hospital and University of Bern, Bern 3010, Switzerland. [23]Graduate School for Cellular and Biomedical Sciences, University of Bern, Bern 3012, Switzerland. [24]Department of Genitourinary Medical Oncology - Research, Division of Cancer Medicine, The University of Texas MD Anderson Cancer Center, Houston, TX 77030, USA. [25]Department of Urology, Icahn School of Medicine at Mount Sinai, New York, NY 10029, USA. [26]Division of Theoretical Bioinformatics, German Cancer Research Center (DKFZ), Heidelberg 69120, Germany. [27]Faculty of Biosciences, Heidelberg University, Heidelberg 69120, Germany. [28]University of Texas MD Anderson Cancer Center, Houston, TX 77030, USA. [29]Korea Advanced Institute of Science and Technology, Daejeon 34141, South Korea. [30]Institute for Research in Biomedicine (IRB Barcelona), The Barcelona Institute of Science and Technology, Barcelona 8003, Spain. [31]Research Program on Biomedical Informatics, Universitat Pompeu Fabra, Barcelona 08002, Spain. [32]Department of Physiology and Biophysics, Weill Cornell Medicine, New York, NY 10065, USA. [33]Institute for Computational Biomedicine, Weill Cornell Medicine, New York, NY 10021, USA. [34]Science for Life Laboratory, Department of Cell and Molecular Biology, Uppsala University, Uppsala SE-75124, Sweden. [35]Division of Applied Bioinformatics, German Cancer Research Center (DKFZ), Heidelberg 69120, Germany. [36]Barcelona Supercomputing Center (BSC), Barcelona 08034, Spain. [37]Queensland Centre for Medical Genomics, Institute for Molecular Bioscience, The University of Queensland, Brisbane, QLD 4072, Australia. [38]CIBIO/InBIO - Research Center in Biodiversity and Genetic Resources, Universidade do Porto, Vairão 4485-601, Portugal. [39]European Molecular Biology Laboratory, European Bioinformatics Institute (EMBL-EBI), Wellcome Genome Campus, Hinxton, Cambridge CB10 1SD, UK. [40]University of Milano Bicocca, Monza 20052, Italy. [41]Peter MacCallum Cancer Centre, University of Melbourne, Melbourne, VIC 3000, Australia. [42]Sir Peter MacCallum Department of Oncology, Peter MacCallum Cancer Centre, University of Melbourne, Melbourne, VIC 3052, Australia. [43]Department of Computer Science, Yale University, New Haven, CT 06520, USA. [44]Department of Molecular Biophysics and Biochemistry, Yale University, New Haven, CT 06520, USA. [45]Program in Computational Biology and Bioinformatics, Yale University, New Haven, CT 06520, USA. [46]Center for Cancer Research, Massachusetts General Hospital, Boston, MA 02129, USA. [47]Department of Pathology, Massachusetts General Hospital, Boston, MA 02115, USA. [48]Bioinformatics Research Centre (BiRC), Aarhus University, Aarhus 8000, Denmark. [49]CNAG-CRG, Centre for Genomic Regulation (CRG), Barcelona Institute of Science and Technology (BIST), Barcelona 08028, Spain. [50]Universitat Pompeu Fabra (UPF), Barcelona 08003, Spain. [51]Biomolecular Engineering Department, University of California Santa Cruz, Santa Cruz, CA 95064, USA. [52]Department of Internal Medicine, Stanford University, Stanford, CA 94305, USA. [53]Massachusetts General Hospital, Boston, MA 02114, USA. [54]Center for Precision Health, School of Biomedical Informatics, The University of Texas Health Science Center, Houston, TX 77030, USA. [55]The Donnelly Centre, University of Toronto, Toronto, ON M5S 3E1, Canada. [56]Health Data Science Unit, University Clinics, Heidelberg 69120, Germany. [57]Institute of Pharmacy and Molecular Biotechnology and BioQuant, Heidelberg University, Heidelberg 69120, Germany. [58]Massachusetts General Hospital Center for Cancer Research, Charlestown, MA 02129, USA. [59]Simon Fraser University, Burnaby, BC V5A 1S6, Canada. [60]Department of Medical Biophysics, University of Toronto, Toronto, ON M5S 1A8, Canada. [61]Computational Biology Center, Memorial Sloan Kettering Cancer Center, New York, NY 10065, USA. [62]Department of Biology, ETH Zurich, Zürich 8093, Switzerland. [63]Department of Computer Science, ETH Zurich, Zurich 8092, Switzerland. [64]SIB Swiss Institute of Bioinformatics, Lausanne 1015, Switzerland. [65]University Hospital Zurich, Zurich 8091, Switzerland. [66]Clinical Bioinformatics, Swiss Institute of Bioinformatics, Geneva 1202, Switzerland. [67]Institute for Pathology and Molecular Pathology, University Hospital Zurich, Zurich 8091, Switzerland. [68]Institute of Molecular Life Sciences, University of Zurich, Zurich 8057, Switzerland. [69]MIT Computer Science and Artificial Intelligence Laboratory, Massachusetts Institute of Technology, Cambridge, MA 02139, USA. [70]Controlled Department and Institution, New York, NY 10065, USA. [71]Englander Institute for Precision Medicine, Weill Cornell Medicine, New York, NY 10065, USA. [72]Research Core Center, National Cancer Centre Korea, Goyang-si 410-769, South Korea. [73]Department of Health Sciences and Technology, Sungkyunkwan University School of Medicine, Seoul 06351, South Korea. [74]Samsung Genome Institute, Samsung Medical Center, Seoul, Korea. [75]Institute of Computer Science, Polish Academy of Sciences, Warsawa 01-248, Poland. [76]Genome Biology Unit, European Molecular Biology Laboratory (EMBL), Heidelberg 69117, Germany. [77]Institute of Biomedicine, Sahlgrenska Academy at University of Gothenburg, Gothenburg, Sweden. [78]Department of Biology, ETH Zurich, Wolfgang-Pauli-Strasse 27, 8093 Zürich, Switzerland. [79]Harvard University, Cambridge, MA 02138, USA. [80]Memorial Sloan Kettering Cancer Center, New York, NY 10065, USA. [81]Department of Information Technology, Ghent University, Ghent B-9000, Belgium. [82]Department of Plant Biotechnology and Bioinformatics, Ghent University, Ghent B-9000, Belgium. [83]Yale School of Medicine, Yale University, New Haven, CT 06520, USA. [84]Institute for Research in Biomedicine (IRB Barcelona), The Barcelona Institute of Science and Technology, Barcelona, Spain. [85]Division of Hematology-Oncology, Samsung Medical Center, Sungkyunkwan University School of Medicine, Seoul 06351, South Korea. [86]Samsung Advanced Institute for Health Sciences and Technology, Sungkyunkwan University School of Medicine, Seoul 06351, South Korea. [87]Cheonan Industry-Academic Collaboration Foundation, Sangmyung University, Cheonan 31066, South Korea. [88]Spanish National Cancer Research Centre, Madrid 28029, Spain. [89]Department of Computer Science, Princeton University, Princeton, NJ 08540, USA. [90]Bern Center for Precision Medicine, University Hospital of Bern, University of Bern, Bern 3008, Switzerland. [91]Englander Institute for Precision Medicine, Weill Cornell Medicine and New York Presbyterian Hospital, New York, NY 10021, USA. [92]Meyer Cancer Center, Weill Cornell Medicine, New York, NY 10065, USA. [93]Pathology and Laboratory, Weill Cornell Medical College, New York, NY 10021, USA. [94]Vall d'Hebron Institute of Oncology: VHIO, Barcelona 08035, Spain. [95]Indiana University, Bloomington, IN 47405, USA. [96]Vancouver Prostate Centre, Vancouver, BC V6H 3Z6, Canada. [97]cBio Center, Dana-Farber Cancer Institute, Harvard Medical

School, Boston, MA 02115, USA. [98]Department of Cell Biology, Harvard Medical School, Boston, MA 02115, USA. [99]Department of Cancer Biology, Dana-Farber Cancer Institute, Boston, MA 02215, USA. [100]cBio Center, Dana-Farber Cancer Institute, Boston, MA 02215, USA. [101]Finsen Laboratory and Biotech Research & Innovation Centre (BRIC), University of Copenhagen, Copenhagen 2200, Denmark. [102]Department of Genetics, Stanford University School of Medicine, Stanford, CA 94305, USA. [103]CREST, Japan Science and Technology Agency, Tokyo 113-0033, Japan. [104]Department of Medical Science Mathematics, Medical Research Institute, Tokyo Medical and Dental University, Bunkyo-kuTokyo 113-8510, Japan. [105]Laboratory for Medical Science Mathematics, Department of Biological Sciences, Graduate School of Science, The University of Tokyo, Bunkyo-kuTokyo 113-0033, Japan. [106]Department of Oncology-Pathology, Science for Life Laboratory, Karolinska Institute, Stockholm, Sweden. [107]Department of Gene Technology, Tallinn University of Technology, Tallinn 12616, Estonia. [108]Genetics & Genome Biology Program, SickKids Research Institute, The Hospital for Sick Children, Toronto, ON M5G 1X8, Canada. [109]Institució Catalana de Recerca i Estudis Avançats (ICREA), Barcelona 08010, Spain. [110]Department of Clinical and Molecular Medicine, Faculty of Medicine and Health Sciences, Norwegian University of Science and Technology, Trondheim 7030, Norway. [111]Department of Information Technology, Ghent University, Interuniversitair Micro-Electronica Centrum (IMEC), Ghent B-9000, Belgium. [112]Science for Life Laboratory, Department of Immunology, Genetics and Pathology, Uppsala University, Uppsala SE-75108, Sweden. [113]School of Computer Science and Technology, Xi'an Jiaotong University, Xi'an 710048, China. [114]School of Electronic and Information Engineering, Xi'an Jiaotong University, Xi'an 710048, China. [115]The McDonnell Genome Institute at Washington University, St Louis, MO 63108, USA. [116]Department of Urology, Charité Universitätsmedizin Berlin, Berlin 10117, Germany. [117]Department of Molecular and Human Genetics, Baylor College of Medicine, Houston, TX 77030, USA. [118]Human Genome Sequencing Center, Baylor College of Medicine, Houston, TX 77030, USA. [119]Oregon Health & Sciences University, Portland, OR 97239, USA. [120]Department of Medicine and Therapeutics, The Chinese University of Hong Kong, Shatin, NTHong Kong, China. [121]The University of Texas Health Science Center at Houston, Houston, TX 77030, USA. [122]Department of Biomedical Informatics, College of Medicine, The Ohio State University, Columbus, OH 43210, USA. [123]The Ohio State University Comprehensive Cancer Center (OSUCCC—James), Columbus, OH 43210, USA. [124]The University of Texas School of Biomedical Informatics (SBMI) at Houston, Houston, TX 77030, USA. [125]Department of Biochemistry and Molecular Genetics, Feinberg School of Medicine, Northwestern University, Chicago, IL 60637, USA. [126]Institute of Molecular Life Sciences and Swiss Institute of Bioinformatics, University of Zurich, Zurich 8057, Switzerland

