## [Peer Review File · Nature Communications]

Reviewers' comments:

Reviewer #1 (Remarks to the Author):

Authors present a computational framework for cancer driver discovery that could be equally well applied to the coding and non-coding genomic regions, a unique and conceptually satisfying feature. The approach to predict a background mutation rate is really impressive.

Using a list of known cancer drivers as a reference, DriverPower algorithms achieves the both the highest precision and recall as compared to other similar algorithms. While not surprising with such a high recall value is, it is somehow disappointing that so few new driver candidates were found. Does this mean that the driver discovery is over? Or that the DriverPower was somehow subtle overtrained on the known drivers set?

Reviewer #2 (Remarks to the Author):

Shuai et al. present their method DriverPower for the computational identification of significantly mutated genomic elements. Their method builds on previous work modeling the background mutation rate across various regions of the genome to identify genes and intergenic elements mutated in more patients than would be expected by chance given the background model. They apply their method to the PCAWG dataset of 2,583 cancer whole genomes and report 217 coding and 95 non-coding driver candidates. Most of these are already known drivers, but some are new. They show that their method has a higher F1-score than other methods.

The manuscript is well written, and the results are solid. The authors have already benefited from a rigorous round of internal reviews within PCAWG, during which many errors were caught and corrected, leaving the current manuscript quite polished. The references to previous work are complete and fair, and the findings are confident.

One apparent typo: in the conclusions (p.9) it is stated, "DriverPower called nearly twice as many non-coding driver events than coding ones." This is reversed: there were 217 coding drivers called,

compared to 95 non-coding. Please correct to "DriverPower called nearly twice as many coding driver events as non-coding ones."

Other than that, the manuscript is ready for publication.

Reviewer #1:

Authors present a computational framework for cancer driver discovery that could be equally well applied to the coding and non-coding genomic regions, a unique and conceptually satisfying feature. The approach to predict a background mutation rate is really impressive.

Using a list of known cancer drivers as a reference, DriverPower algorithms achieves the both the highest precision and recall as compared to other similar algorithms. While not surprising with such as high recall value is, it is somehow disappointing that so few new driver candidates were found. Does this mean that the driver discovery is over? Or that the DriverPower was somehow subtle overtrained on the known drivers set?

In fact, DriverPower did not have the highest recall among the algorithms tested, but provides the best balance between recall and precision (F1-score). This suggests that DriverPower can find a higher fraction of candidates not covered by reference driver lists than other methods including MutSig and ncdDetect (see Fig. 2b,d).

We have carefully calibrated DriverPower to avoid overfitting. We used randomized elements that avoid overlapping any test elements in order to train the background mutation rate model. Parameter tuning was conducted with independent training and test tumour sets. Moreover, three different randomized mutation sets were also used to study the distribution of p-values under the null hypothesis and the results suggested no p-value inflation or deflation.

The ratio of coding to non-coding driver mutations discovered by DriverPower is roughly 2:1. Most (82.5%, or 179/217 in COSMIC Cancer Gene Census) coding driver candidates were not novel, as we would expect. However, many of the non-coding driver candidates that DriverPower identified are indeed novel. Some novel candidates are supported by orthogonal evidence such as the CDS of EEF1A2 and MEF2B, the splice site of SGK1 and the intronic region of GPR126, were described in our manuscript. Other novel non-coding candidates discovered by DriverPower and contributed to the PCAWG-consensus set are discussed in another Pan-Cancer Analysis of Whole Genomes (PCAWG) paper (<https://doi.org/10.1101/237313>; reference added to the manuscript).

The main surprise is that novel non-coding cancer drivers are much rarer than we had expected. We have summarized several possible reasons for this observation in the manuscript (lines 313-329). A more detailed discussion is also available at the aforementioned PCAWG paper (<https://doi.org/10.1101/237313>). If a novel driver is missed due to technical reasons such as insufficient sample size and/or read coverage, we expect to identify them with larger dataset and/or deeper sequencing in the future. In fact, many (13/38) tumour types have <30 samples (note added to line 322) in the PCAWG dataset and it's estimated that around 216 mutations in the TERT promoter have been missed in the PCAWG dataset due to coverage issues (<https://doi.org/10.1101/237313>; added to line 318).

Most importantly, just 3.7% of the genome is covered by annotated coding and non-coding functional elements, and it is these regions that were interrogated by DriverPower in order to avoid loss of statistical power. There may very well be novel drivers outside of the studied regions, in functional elements that have not yet been annotated. In summary, it is still too early

to claim that driver discovery is over. In response to the reviewer's comment, we have added a discussion of these issues in Discussion (lines 324-327).

Reviewer #2:

Shuai et al. present their method DriverPower for the computational identification of significantly mutated genomic elements. Their method builds on previous work modeling the background mutation rate across various regions of the genome to identify genes and intergenic elements mutated in more patients than would be expected by chance given the background model. They apply their method to the PCAWG dataset of 2,583 cancer whole genomes and report 217 coding and 95 non-coding driver candidates. Most of these are already known drivers, but some are new. They show that their method has a higher F1-score than other methods.

The manuscript is well written, and the results are solid. The authors have already benefited from a rigorous round of internal reviews within PCAWG, during which many errors were caught and corrected, leaving the current manuscript quite polished. The references to previous work are complete and fair, and the findings are confident.

One apparent typo: in the conclusions (p.9) it is stated, "DriverPower called nearly twice as many non-coding driver events than coding ones." This is reversed: there were 217 coding drivers called, compared to 95 non-coding. Please correct to "DriverPower called nearly twice as many coding driver events as non-coding ones."

This has been corrected in the manuscript (lines 313-314).

Other than that, the manuscript is ready for publication.

We hope so too!

REVIEWERS' COMMENTS:

Reviewer #1 (Remarks to the Author):

I'm satisfied with authors response and see no reason to delay publication